# Safety and clinical impact of adenosine stress perfusion cardiac magnetic resonance in Asian patients with known or suspected coronary artery disease

**Yodying Kaolawanich[1], Thammarak Songsangjinda[1,2], Kanchalaporn Jirataiporn[1], Ahthit Yindeengam[1], Rungroj Krittayaphong[1]** *

**1** Division of Cardiology, Department of Medicine, Faculty of Medicine Siriraj Hospital, Mahidol University Bangkok, Thailand, **2** Division of Cardiology, Department of Medicine, Faculty of Medicine, Prince of Songkla University, Songkhla, Thailand

* rungroj.kri@mahidol.ac.th

**Data Availability Statement:** Data from this study are available upon reasonable request. However, the request needs to submit a short document with

## Abstract

### Background

Adenosine stress cardiac magnetic resonance (CMR) imaging is increasingly utilized for evaluating patients with known or suspected coronary artery disease (CAD). This study aims to assess the safety and clinical impact of adenosine stress CMR in a tertiary care setting in Thailand.

### Methods

A total of 3,768 consecutive patients aged 18 years and above who underwent adenosine stress CMR between 2017 and 2020 were included in the study. Patient records were reviewed to collect data on clinical characteristics, hemodynamic measurements, complications during or immediately after CMR, and the rates of clinical changes resulting from CMR.

### Results

Among the included patients, the primary indications for adenosine stress CMR were risk stratification in suspected CAD (70.8%) and the assessment of myocardial ischemia/viability in patients with known CAD (26.5%). There were no reported deaths or acute myocardial infarctions during the procedure. Major complications, specifically acute pulmonary edema requiring hospital observation or admission for further management, occurred in four patients (0.11%), all of whom were elderly (ranging from 75 to 91 years) with a history of heart failure. Non-major complications were observed in 13.7% of patients, with dyspnea (9.8%) and mild chest pain (5.6%) being the most common. CMR provided a completely new diagnosis in 26.2% of patients. Overall, stress CMR resulted in a change in diagnosis or management for 48% of patients.

description specify the reason for the request. The data belong to the Cardiovascular Imaging unit of Siriraj Hospital. Request to use or access the data has to be considered by Cardiovascular Imaging staff whether the request is reasonable. If reasonable, data sharing agreement may be needed. Besides even we remove the ID of patients in the datafile, the remaining data sometime can track or identify the patient. For data inquiries please contact the data manager: Poom Sairat, M. S. e-mail: poom.kaab@gmail.com.

**Funding:** The author(s) received no specific funding for this work.

**Competing interests:** The authors have declared that no competing interests exist.

**Abbreviations:** CAD, coronary artery disease; CMR, cardiac magnetic resonance; ECG, electrocardiography; LGE, late gadolinium enhancement; MI, myocardial infarction; NYHA, New York Heart Association; SD, standard deviation.

## Conclusion

Adenosine stress CMR was found to be safe and to have a significant impact on clinical management in Asian patients with known or suspected CAD. These findings support the use of adenosine stress CMR as a valuable tool for evaluating and guiding treatment decisions in this patient population.

## Introduction

Coronary artery disease (CAD) is a significant global health burden, with a high prevalence and impact on individuals and healthcare systems. Accurate diagnosis and risk stratification of patients with known or suspected CAD are of utmost importance. Stress cardiac magnetic resonance (CMR), commonly using adenosine, is increasingly being utilized. CMR provides a comprehensive assessment of CAD with very high accuracy [1]. It can assess global and regional ventricular function, myocardial ischemia, and infarction in a single study. Moreover, stress CMR offers strong evidence for prognosis, including mortality, in patients with known or suspected CAD [2, 3].

Stress CMR has demonstrated a significant impact on the diagnosis and management of a large patient population, as evidenced by The EuroCMR registry, which included more than 9,500 patients suspected of CAD or suspected ischemia in known CAD who underwent stress tests [4]. Previous studies have reported that stress CMR resulted in a substantial alteration in clinical care for approximately 60–70% of patients [4, 5]. The safety profile of stress CMR has been documented in prior studies. Menadas et al. demonstrated that dipyridamole stress CMR was feasible and safe, with a rate of severe immediate complications of 0.06% [6]. Bruder et al. also showed a very low rate of severe complications of stress CMR using adenosine and dobutamine, with a rate of 0.026% [4]. However, there are limited studies regarding the clinical impact and safety of stress CMR in Asia, and most of them were relatively small in scale [7, 8].

This study aims to assess the safety and clinical impact of adenosine stress CMR in a tertiary care setting in Thailand.

## Methods

### Study population

This was a retrospective observational study conducted at Siriraj Hospital, an academic medical center in Bangkok, Thailand. We included consecutive patients aged 18 years or older who were referred for adenosine stress CMR for clinical purposes during 2017 to 2019. Patients were excluded if they had incomplete CMR scans or if they did not have follow-up data after CMR. Information on baseline demographic variables was obtained from the electronic medical record. Hypertension was defined as a self-reported history of hypertension, the use of anti-hypertensive medication, or an office blood pressure of ≥140/90 mmHg. Diabetes was defined as a self-reported history of diabetes and/or receiving anti-diabetic treatment, or a fasting glucose level of ≥126 mg/dL. Dyslipidemia was defined as a total cholesterol level of ≥240 mg/dL, a low-density lipoprotein (LDL) cholesterol level of ≥130 mg/dL, a high-density lipoprotein (HDL) cholesterol level of <40 mg/dL, a triglyceride level of ≥200 mg/dL, and/or treatment with a lipid-lowering agent. The protocol for this study was approved by the Siriraj Institutional Review Board. The Ethics Committee waived the requirement of written informed

consent for participation due to the retrospective design of the study. Data were accessed for research purposes between 1st April 2020 and 30th January 2023.

## CMR protocol and analysis

CMR was performed using standardized protocols recommended by the Society of Cardiovascular Magnetic Resonance (SCMR) [9] on a 3T scanner (Philips Medical Systems, Best, The Netherlands) and interpreted by experienced readers. The CMR protocol included ventricular function assessment using a standard steady-state free precession sequence obtained in short and long-axis views, myocardial first-pass perfusion, and late gadolinium enhancement (LGE). Three short-axis slices at the apical, mid, and basal left ventricular levels were chosen for perfusion imaging with an electrocardiographic (ECG)-triggered, T1-weighted, inversion recovery single-shot turbo gradient echo sequence. Typical image parameters were described elsewhere [10].

Venous access was obtained in both upper limbs, with one being used for continuous adenosine infusion and the other being used for gadolinium contrast infusion at peak stress. Patients were continuously monitored with peripheral oxygen saturation probe, heart rate, and real-time electrocardiography (ECG) throughout the CMR scan. Blood pressure (BP) was recorded before starting the infusion and was checked every minute during adenosine infusion. The myocardial first-pass perfusion study was performed by injecting 0.05 mmol/kg of gadolinium contrast agent (Magnevist, Bayer Schering Pharma, Berlin, Germany) at a rate of 4 mL/s immediately after a 4-minute infusion of 140 mcg/kg/min of adenosine [9]. Patients were asked about any symptoms experienced during the infusion in order to assess their hemodynamic response and monitor for any potential complications. The adenosine infusion was discontinued prematurely if requested by the patient or in the presence of progressive or severe angina, dyspnea, a decrease in systolic pressure >40 mmHg, severe arrhythmias, or other adverse effects. LGE imaging obtained 10–15 minutes after administration of intravenous gadolinium (0.15 mmol/kg), as per published guidelines [9]. CMR images were interpreted by standard methods [11]. In brief, perfusion images were read, and each of the 16 segments was visualized (segment-17 at the apex was not visualized). Inducible ischemia was defined as a subendocardial perfusion defect that (i) persisted beyond peak myocardial enhancement and for several RR intervals, (ii) was more than two pixels wide, (iii) followed one or more coronary arteries, and (iv) showed absence of LGE in the same segment. Dark-banding artefacts were recorded if an endocardial dark band appeared at the arrival of contrast in the LV cavity before contrast arrival in the myocardium. LGE images were also analyzed using visual assessment. LGE was considered present only if confirmed on both the short-axis and at least one other orthogonal plane. CMR diagnosis of CAD includes either a stress-inducible perfusion defect or the presence of ischemic LGE. The CMR diagnosis of nonischemic cardiomyopathy includes a nonischemic LGE pattern (e.g., midwall LGE for dilated cardiomyopathy) without stress-induced perfusion defects.

## Complications of CMR

Major complications were defined as death, resuscitation, or any other condition related to the CMR procedure that required monitoring as an inpatient for at least 1 night after the CMR scan (e.g., acute myocardial infraction, acute pulmonary edema, ischemic stroke, arrhythmias, and so on). Non-major complications were defined as any complications related to CMR that did not fulfill the criteria for severe complications (e.g., dyspnea, chest pain, allergic reactions without shock, problems related to intravenous lines, and so forth).

### Influence of CMR on subsequent clinical management

Two cardiologists, who were blinded to the CMR results, reviewed patient information, including baseline characteristics, indications for CMR, adverse events during CMR studies, and changes in management after CMR. They independently assessed the clinical impact of each stress CMR by reviewing electronic medical records up to the next outpatient visit with the ordering provider. A "completely new diagnosis" was defined as a diagnosis occurring only if it was previously unknown to the referring physician.

### Statistical analysis

Statistical analyses were performed using IBM SPSS Statistics for Windows, version 20.0 (IBM Corp., Armonk, NY, USA). Continuous variables with a normal distribution were presented as mean ± standard deviation (SD), and continuous variables with a non-normal distribution were presented as median and interquartile range. The normality distribution of the variables was examined using the Kolmogorov-Smirnov test. Categorical variables were presented as absolute numbers and percentages. Differences were compared using Student's unpaired t-test or Mann-Whitney U test for continuous variables, and the chi-square test or Fisher's exact test for categorical variables, as appropriate. All statistical tests were two-tailed, and p-values less than 0.05 were considered to indicate statistical significance.

## Results

### Patient characteristics

A total of 3,785 patients were scanned within a period of 31 months. Flow diagram is presented in **Fig 1**. Ten patients with a contraindication to adenosine stress CMR (e.g., severe claustrophobia, known or suspected bronchoconstrictive or bronchospastic disease) and 7 patients with incomplete CMR scans were excluded. As a result, 3,768 patients were included in the final analysis. The main indications for adenosine stress CMR were risk stratification in suspected CAD (70.8%) and the assessment of myocardial ischemia/viability in patients with known CAD (26.5%) (**Fig 1**).

Clinical characteristics of the patients are presented in **Table 1**. The mean age was 67.5 ±13.6 years, and 1,818 (48.2%) were male. Hypertension was the most prevalent CAD risk factor, observed in 73.9% of the patients. A total of 1,448 (38.4%) had diabetes mellitus. Among the patients, 1,082 had known CAD, with 264 having a prior myocardial infarction (MI). Forty-six percent of the patients presented with dyspnea, while twenty-two percent presented with chest pain.

### Hemodynamic measurements

An increase in heart rate was observed in 2,973 (79.0%) patients, and a reduction in blood pressure was observed in 2,987 (79.5%) patients. **Fig 2** depicts the hemodynamic effects of adenosine in our patients. Overall, there was a significant decrease in mean systolic and diastolic blood pressure (systolic: 135.2 ± 19.9 [rest] versus 124.8 ± 19.4 mmHg [during adenosine infusion], $p < 0.001$; diastolic: 74.7 ± 13.6 [rest] versus 66.0 ± 12.7 mmHg [during adenosine infusion], $p < 0.001$, respectively), accompanied by a compensatory increase in mean heart rate (75.7 ± 13.9 [rest] versus 83.4 ± 14.3 beats per minute [during adenosine infusion], $p < 0.001$).

### Major and non-major complications

Major complications occurred in four patients (0.11%), all of whom had acute pulmonary edema requiring hospital observation or admission for further management. All four patients,

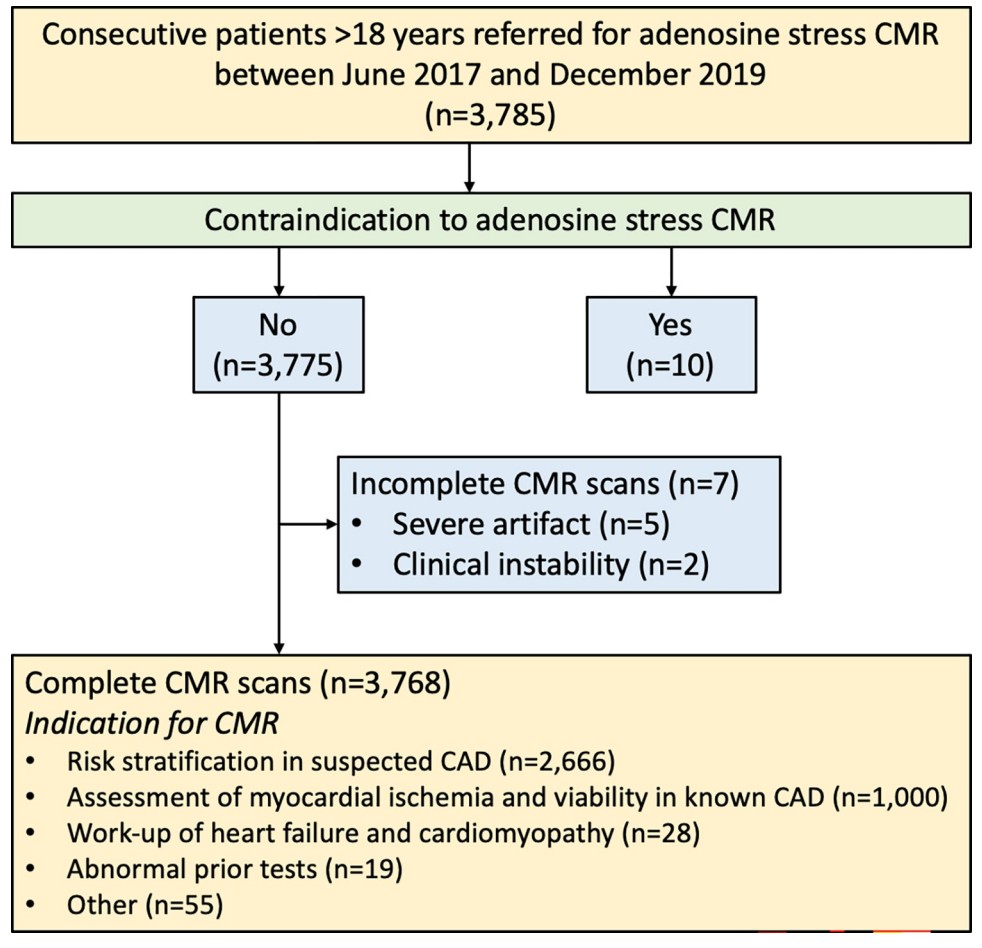

**Fig 1. Flow diagram of the study. Abbreviations:** CAD = coronary artery disease, CMR = cardiac magnetic resonance.

aged over 75 (ranging from 75 to 91 years), required hospital observation or admission for further management. They had previously been diagnosed with heart failure, classified as New York Heart Association (NYHA) class II-III, with two having reduced left ventricular ejection fraction and two having preserved left ventricular ejection fraction. **Table 2** shows major complications during or immediately after CMR scans. There was no reported death or acute MI during the procedure.

Non-major complications occurred in 517 patients (13.7%), with 368 experiencing dyspnea (9.8%) and 209 reporting mild chest pain (5.6%) as the most common symptoms. The remaining patients experienced nausea (n = 7), local complications at intravenous access sites, such as small hematoma, edema, or phlebitis (n = 7), and contrast allergy without shock (n = 3). No patients exhibited atrioventricular block. Among patients with dyspnea, there was no significant difference between those with a history of heart failure (n = 34/447; 7.6%) and those without (n = 334/3321; 10.1%), p = 0.10. Similarly, among patients with mild chest pain, there was no significant difference between those with known CAD (n = 65/1082; 6.0%) and those with suspected CAD (n = 114/2686; 5.4%), p = 0.43. Noted, there is no difference in the rate of patients with positive stress CMR who developed dyspnea or mild chest pain during CMR (p>0.05 for both). **Fig 3** demonstrates non-major complications during or immediately after CMR scans.

**Table 1. Clinical characteristics of the study population.**

| Characteristics | Total (n = 3,768) |
|---|---|
| Age (years) | 67.5±13.6 |
| Male | 1,818 (48.2) |
| Body mass index (kg/m$^2$) | 25.5±4.6 |
| CAD risk factors | |
| Hypertension | 2,785 (73.9) |
| Hyperlipidemia | 2,401 (63.7) |
| Diabetes mellitus | 1,448 (38.4) |
| Cigarette smoking | 115 (3.1) |
| Family history of CAD | 23 (0.6) |
| Medical history | |
| Known CAD | 1,082 (28.7) |
| Prior myocardial infarction | 264 (7.0) |
| Revascularization, PCI | 469 (12.4) |
| Revascularization, CABG | 256 (6.8) |
| History of heart failure | 447 (11.9) |
| Atrial fibrillation | 392 (10.4) |
| Stroke | 251 (6.7) |
| Symptoms | |
| Chest pain | 832 (22.1) |
| Dyspnea | 1,755 (46.6) |
| Other symptoms | 24 (0.6) |
| No symptom | 296 (7.9) |
| Medications | |
| Antiplatelet | 2,025 (53.7) |
| Anticoagulant | 303 (8.0) |
| ACEI or ARB | 1,480 (39.3) |
| Beta blocker | 1,977 (52.5) |
| Calcium channel blocker | 1,100 (29.2) |
| Diuretic | 744 (19.7) |
| Nitrate | 746 (19.8) |
| Oral hypoglycemic agent | 891 (23.6) |
| Insulin | 204 (5.4) |

Values are mean ± standard deviation or number (%).

**Abbreviations:** ACEI = angiotensin-converting enzyme inhibitors, ARB = angiotensin receptor blockers, CABG = coronary artery bypass graft, CAD = coronary artery disease, PCI = percutaneous coronary intervention.

## Clinical impact of CMR

**Table 3** demonstrates the number of patients with a completely new diagnosis and changes in management after CMR. Adenosine stress CMR provided a diagnosis of 650 cases of CAD in patients with unknown CAD (before CMR). Additionally, comprehensive CMR examinations enabled the identification of nonischemic cardiomyopathies, such as dilated cardiomyopathy, hypertrophic cardiomyopathy, and myocarditis. CMR also facilitated the diagnosis of severe valvular disease, necessitating valvular surgery or intervention.

CMR resulted in therapeutic consequences in 1,110 (29.5%) patients, including changes in medication in 21.4% and invasive procedures in 14.2%. Overall, CMR had an impact on the

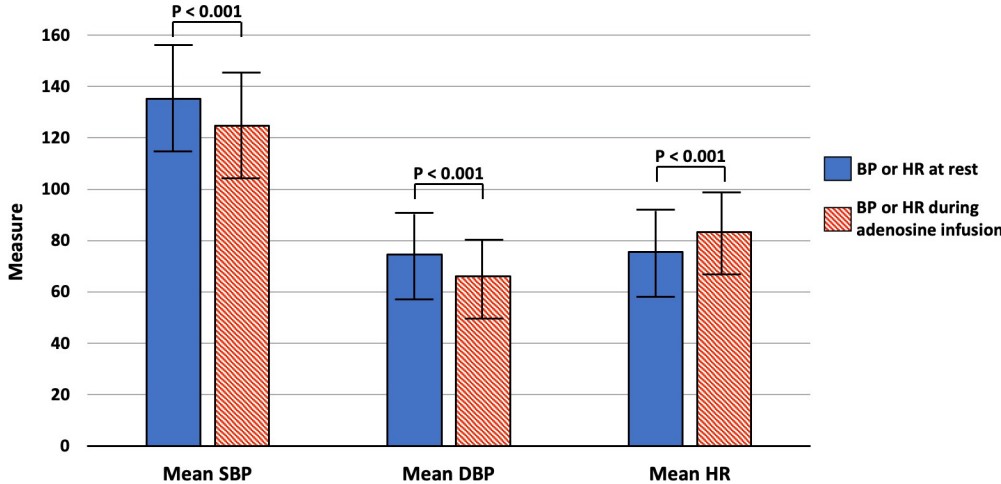

**Fig 2. Hemodynamic parameters at rest and during adenosine infusion. Abbreviations**: DBP = diastolic blood pressure, HR = heart rate, SBP = systolic blood pressure.

diagnosis and patient management (completely new diagnosis and/or therapeutic consequences) in 48% of the patients.

## Discussion

In this study, we have shown the main following findings: 1) Adenosine stress CMR was found to be safe, with a very low rate of major complications, specifically acute pulmonary edema (0.11%). All cases of acute pulmonary edema occurred in elderly patients with a history of heart failure. 2) Non-major complications were observed in 13.6% of the patients, with the most common being dyspnea and mild chest pain. These symptoms were not associated with patient characteristics such as heart failure or CAD status. 3) Adenosine stress CMR had a significant impact on diagnosis and clinical management, resulting in 48% of patients receiving a new diagnosis or experiencing changes in their management.

Stress CMR has emerged as a prominent imaging modality for the detection and risk stratification of patients with known or suspected CAD. CMR can provide integrated information

**Table 2. Major complications during or immediately after CMR scans.**

|  | Total (n = 3,768) |
|---|---|
| Major complications | 4 (0.11) |
| Acute pulmonary edema | 4 (0.11) |
| Acute myocardial infraction | 0 |
| [a]Unstable angina | 0 |
| Cardiac arrest | 0 |
| Sustained ventricular tachycardia | 0 |
| Ischemic stroke or transient ischemic attack | 0 |
| Severe symptomatic hypotension | 0 |
| Anaphylactic shock | 0 |

Values are number (%).

**Abbreviations:** CMR = cardiac magnetic resonance.

[a]Chest pain of >20 min duration despite treatment, requiring hospital admission.

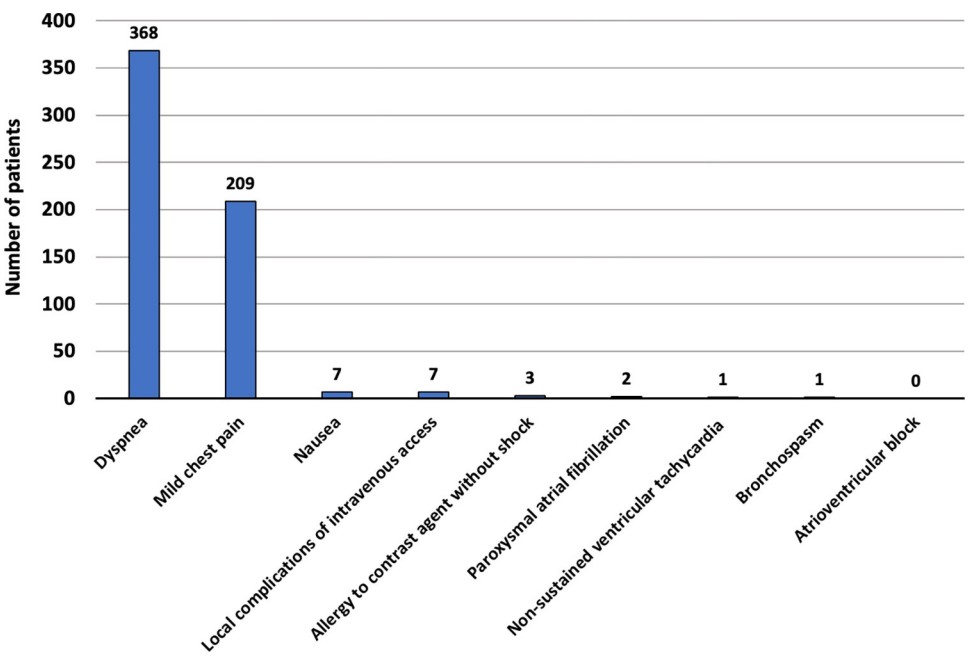

**Fig 3. Non-major complications during or immediately after CMR scans.**

of cardiac function, structural changes such as LGE, and myocardial ischemia by studying myocardial perfusion in one examination [12]. Stress CMR, whether using vasodilators or dobutamine, has strong evidence for the diagnosis and prognosis of patients with known or suspected CAD [1–3]. Dipyridamole, one of the vasodilators used for stress CMR, has been shown to be safe. Menadas et al. reported a very low rate of severe immediate complications and demonstrated that inducible ischemia was the only factor identified as being associated with serious complications [6]. Adenosine has been the most commonly used vasodilator for stress CMR and has been shown to exhibit better sensitivity and specificity compared to dipyr- idamole [13]. Our center has demonstrated good accuracy with adenosine stress CMR, achiev- ing a sensitivity of 89.5% and a specificity of 78.6% compared to invasive coronary angiography [14]. These results are consistent with other published papers that also demon- strate good accuracy [13].

Adenosine stress CMR has also demontrated a favorable safety profile in several studies conducted in Western countries [4, 15]. The safety of adenosine stress CMR in Asia has been studied by Raj et al. [8] in 1,057 patients and Tsang et al. [7] in 98 patients. Both studies dem- onstrated a very low rate (<0.5%) of adverse events during or immediately after CMR. Our study, conducted with a larger patient population of 3,768 patients, also yielded consistent results. The majority of our patients exhibited a hemodynamic response to adenosine, charac- terized by a decrease in blood pressure and an increase in heart rate. There were no reported deaths or acute MIs during or immediately after CMR. Four patients with acute pulmonary edema were reported, all of whom were elderly and had symptomatic heart failure. It is known that a comprehensive stress CMR protocol requires patient cooperation, as they have to remain supine in a magnet for at least 30 minutes and repeatedly hold their breath. Patients with symptomatic heart failure, especially those with fluid retention, are at a high risk for a heart failure event during or after CMR. A study by Raj et al. in India demonstrated that three patients experienced severe breathlessness during adenosine infusion and required further management, and one of them had a reduced LVEF [8]. This was consistent with our results.

**Table 3. Impact of CMR on diagnosis and patient management.**

| Variables | Total (n = 3,768) |
|---|---|
| Completely new diagnosis | 986 (26.2%) |
| CAD | 650 |
| Non-CAD | 336 |
| Dilated cardiomyopathy | 93 |
| Hypertrophic cardiomyopathy | 55 |
| Significant valvular disease | 53 |
| Vascular disease | 35 |
| Hypertensive heart disease | 19 |
| Myocarditis | 12 |
| Noncompaction cardiomyopathy | 11 |
| Others | 58 |
| Therapeutic consequences | 1,110 (29.5%) |
| Change in medication | 807 |
| Add new medication | 459 |
| Aspirin | 147 |
| Clopidogrel | 127 |
| Anticoagulation | 31 |
| Beta blocker | 147 |
| Calcium channel blocker | 43 |
| Nitrate | 88 |
| Statin | 100 |
| Discontinued Medication | 201 |
| Dose changed | 249 |
| nvasive procedure | 534 |
| Impact on patient management (completely new diagnosis and/or therapeutic consequences) | 1,814 (48.1%) |

Values are number (%).

**Abbreviations:** CAD = coronary artery disease.

The studies by Bruder and Menadas from Western countries showed differences compared to our data [4, 6]. Bruder et al. reported a very low rate of severe complications (0.026%) from the EuroCMR registry, independent of patient gender or age [4]. Similarly, Menadas et al. reported a very low rate of severe complications (0.08%), with no association found between complications and demographic, clinical, hemodynamic, or CMR-derived parameters [6]. Although our results differed from those of Bruder and Menadas, these discrepancies could be due to chance given the very low event rate. It's worth noting that our study included older patients with a higher prevalence of heart failure, more than double that of Menadas. Non-major complications occurred in 13.7% of our patients, with the most common being dyspnea and mild chest pain. These two symptoms were not related to patient characteristics such as heart failure symptom or CAD status. It is likely that they were effects of adenosine itself, which were not significant and could self-recover without requiring any specific treatment. Overall, we believe that adenosine stress CMR is safe for the majority of referred patients.

Several studies have reported a significant clinical impact of stress CMR. In our study, 48% of patients had a management impact, resulting in new diagnoses or changes in treatment based on CMR results. Approximately 10% of these patients had non-CAD diagnoses such as dilated cardiomyopathy, hypertrophic cardiomyopathy, or severe valvular disease,

requiring different treatments including surgery or interventional procedures. These findings underscore the value of stress CMR. It is important to note that our results may differ from previous studies due to variations in endpoint definitions. For instance, McGraw et al. reported that stress CMR led to active changes in clinical care in about 70% of patients [5]. However, their study included subspecialty consultations, preoperative clearances, or discharge from the Cardiology Clinic as criteria for active clinical change. On the other hand, Bruder et al., whose definition was more similar to ours, found that approximately 60% of patients had an impact from CMR results [4]. It is worth mentioning that Bruder's study included both stress and non-stress CMR. Nonetheless, stress CMR continues to have a significant impact on clinical decision-making, as observed in our study.

Table 4 presents a summary of patients' profiles, complications, and the clinical impact of stress CMR in our study, compared with previously published data [4–8]. Our data demonstrated some degree of similarity in patient profiles, such as having 28% with known CAD, consistent with Menadas et al. and McGraw et al. [5, 6] Notably, our study had a lower proportion of male participants compared to most studies, as male patients may have a higher pretest probability of obstructive CAD and may undergo invasive coronary angiography rather than CMR. Regarding complications, our study and all studies had quite similar very low rates of major complications (<0.3% in all studies), showing a consistent safety profile of stress CMR, while there were some differences in baseline characteristics. As for the clinical impact of CMR, although there were differences in the definitions of impact and/or therapeutic consequences, all studies also showed a consistent and significant impact of CMR. Overall, we have added data from Asia showing that CMR is safe and has a clinical impact compared to studies from Western countries.

For applicability, our data highlights the safety and clinical impact of adenosine stress CMR for patient management. This data will assist clinicians in ensuring that adenosine CMR is indeed very safe and has a clinical impact on patient management. This will also enhance the confidence of clinicians and their patients in the benefits of CMR. Such confidence will promote CMR as a one-stop service offering accurate diagnostic testing with an excellent safety profile and significant clinical impact.

Our study had several limitations. Firstly, the retrospective nature of the study could introduce confounders that cannot be completely eliminated, and reviewing the medical records could potentially introduce some bias. However, the reviewers who assessed the patient information were blinded to the CMR results, which was the best approach we could take. Secondly, it was conducted in a single tertiary center in Thailand, which may limit the generalizability of our findings to different regions. Additionally, the specific definition used in our study may differ from that of others, which could impact the comparability of results. However, our study contributes valuable data from an Asian country, which was previously scarce in this field. Thirdly, certain complications such as bradyarrhythmia, atrioventricular block, or non-sustained ventricular tachycardia might not have been accurately recorded since we were unable to obtain 12-lead ECG recordings during CMR. However, this is a common limitation in many CMR studies, and it's important to note that no patients in our study required treatment for any arrhythmic events. Fourthly, due to the very low rate of events, we were unable to provide a reliable predictor for major complications. The limited number of events hindered our ability to perform a thorough and accurate analysis in this regard. Finally, our study did not include a cost-effective analysis of CMR compared to invasive coronary angiography or nuclear studies, as this was not our primary aim. However, this could serve as an opportunity for future research on this matter.

**Table 4. Summary of patients' characteristics, complications, and clinical impact of stress CMR in published articles.**

| | Kaolawanich, et al. | Menadas, et al. [6] | Bruder, et al. [4] | McGraw, et al. [5] | Raj, et al. [8] | Tsang, et al. [7] |
|---|---|---|---|---|---|---|
| Year of enrollment | 2017–2020 | 2004–2014 | Until 2012 | N/A | 2018–2019 | 2013 |
| Number of patients | 3,768 | 11,984 | 27,781 | 350 | 1,057 | 98 |
| Country | Thailand | Spain | Europe | USA | India | China |
| Asian | 100% | 0% | 0% | 0% | 100% | 100% |
| % with stress test | 100% | 100% | 37.4% | 100% | 100% | 100% |
| Stressors | Adenosine | • Dipyridamole (95.4%) • Dobutamine (4.6%) | • Adenosine (78.3%) • Dobutamine (21.7%) | Regadenoson | Adenosine | Adenosine |
| Age (mean) | 67.5±13.6 | 64±12 | 60 | 59±13.7 | 55.5±9.9 | 64.0±11.4 |
| Male | 48.2% | 60.4% | 65.5% | 46.3% | 87.6% | 71.4% |
| Hypertension | 73.9% | 64.2% | N/A | 74.9% | N/A | N/A |
| Hyperlipidemia | 63.7% | 53.1% | N/A | 53.4% | N/A | N/A |
| Diabetes mellitus | 38.4% | 26.2% | N/A | 34.9% | N/A | N/A |
| Cigarette smoking | 3.1% | 18.0% | N/A | 18.9% | N/A | N/A |
| Family history of CAD | 0.6% | 6.8% | N/A | N/A | N/A | N/A |
| Known CAD | 28.7% | 29.3% | N/A | 31.4% | 94.8% | 52.0% |
| History of MI | 7.0% | 17.1% | N/A | N/A | N/A | N/A |
| History of heart failure | 11.9% | 4.9% | N/A | N/A | N/A | N/A |
| Most common CMR indication | Risk stratification in suspected CAD (70.8%) | N/A | Risk stratification in suspected CAD/Ischemia in known CAD (34.2%) | N/A | Known CAD | Risk stratification in suspected CAD (52.0%) |
| **Complications during or immediately after stress CMR** | | | | | | |
| Death or acute MI | 0 | 0 | 0 | N/A | 0 | 0 |
| Major complication | 4 (0.11%) • Acute pulmonary edema (n = 4) | 10 (0.08%) • Unstable angina (n = 2) • Acute pulmonary edema (n = 2) • Sustained VT (n = 1) • Persistent AF (n = 2) • Asystole (n•1) • TIA (n = 1) • Anaphylactic shock (n = 1) | 7 (0.07%) • Non-sustained VT (n = 2) • Ventricular fibrillation (n = 1) • Overt heart failure (n = 2) • Unstable angina (n = 1) • Anaphylactic shock (n = 1) | N/A | 3 (0.28%) • Unstable angina required hospital admission (n = 3) | 0 |
| Non-major complications/ symptoms | Non-major complications 13.7% | • Non-major complications 1.5% • Minor symptoms 24.8% | Mild complications 3.6% | N/A | • Transient hypotension 1.8% • Severe chest pain 0.5% • Severe breathlessness 0.9% | Adverse effects 63.3% |
| **Clinical impact of stress CMR** | | | | | | |
| New/change diagnosis | 26.2% | N/A | 8.1%[b] | N/A | N/A | N/A |
| Therapeutic consequences | | | | | | |
| • Change in medication | 21.4% | N/A | 24.3% | 18.3% | N/A | N/A |
| • Invasive procedure | 14.2% | N/A | 23.1% | 13.1%[c] | N/A | N/A |

*(Continued)*

**Table 4.** (Continued)

|  | Kaolawanich, et al. | Menadas, et al. [6] | Bruder, et al. [4] | McGraw, et al. [5] | Raj, et al. [8] | Tsang, et al. [7] |
|---|---|---|---|---|---|---|
| Impact on patient management[a] | 48.1% | N/A | 71.4% | 69.5% [d] | N/A | N/A |

[a]Diagnosis and/or therapeutic consequences.

[b]Completely new diagnosis not suspected before.

[c]Angiography with and without revascularization.

[d]Active change in clinical care.

**Abbreviations:** AF = atrial fibrillation, CAD = coronary artery disease, CMR = cardiac magnetic resonance, MI = myocardial infarction, VT = ventricular tachycardia, TIA = transiebnt ischemic attack.

## Conclusions

This study, the largest in Asia to date, aimed to demonstrate the safety profiles and clinical impact of adenosine stress CMR in Asian patients with known or suspected CAD. The findings of this study confirm that adenosine stress CMR is not only safe but also has a significant impact on clinical management in this patient population. These results provide strong support for the utilization of adenosine stress CMR as a valuable tool for evaluating and guiding treatment decisions in patients with known or suspected CAD.

## Author Contributions

**Conceptualization:** Yodying Kaolawanich, Thammarak Songsangjinda, Rungroj Krittayaphong.

**Data curation:** Yodying Kaolawanich, Thammarak Songsangjinda, Kanchalaporn Jirataiporn, Ahthit Yindeengam, Rungroj Krittayaphong.

**Formal analysis:** Yodying Kaolawanich, Thammarak Songsangjinda, Kanchalaporn Jirataiporn, Ahthit Yindeengam, Rungroj Krittayaphong.

**Investigation:** Yodying Kaolawanich, Thammarak Songsangjinda, Kanchalaporn Jirataiporn, Ahthit Yindeengam, Rungroj Krittayaphong.

**Methodology:** Yodying Kaolawanich, Thammarak Songsangjinda, Rungroj Krittayaphong.

**Project administration:** Yodying Kaolawanich, Rungroj Krittayaphong.

**Resources:** Yodying Kaolawanich, Thammarak Songsangjinda, Kanchalaporn Jirataiporn, Ahthit Yindeengam, Rungroj Krittayaphong.

**Software:** Yodying Kaolawanich, Thammarak Songsangjinda, Rungroj Krittayaphong.

**Supervision:** Yodying Kaolawanich, Thammarak Songsangjinda, Rungroj Krittayaphong.

**Validation:** Yodying Kaolawanich, Thammarak Songsangjinda, Rungroj Krittayaphong.

**Visualization:** Yodying Kaolawanich, Thammarak Songsangjinda, Rungroj Krittayaphong.

**Writing – original draft:** Yodying Kaolawanich, Thammarak Songsangjinda, Rungroj Krittayaphong.

**Writing – review & editing:** Yodying Kaolawanich, Thammarak Songsangjinda, Kanchalaporn Jirataiporn, Ahthit Yindeengam, Rungroj Krittayaphong.

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
