## [Decision Letter · Decision Letter 0]

10 Sep 2023

PONE-D-23-23709Safety and Clinical Impact of Adenosine Stress Perfusion Cardiac Magnetic Resonance in Asian Patients with Known or Suspected Coronary Artery DiseasePLOS ONE

Dear Dr. Krittayaphong,

Thank you for submitting your manuscript to PLOS ONE. After careful consideration, we feel that it has merit but does not fully meet PLOS ONE’s publication criteria as it currently stands. Therefore, we invite you to submit a revised version of the manuscript that addresses the points raised during the review process.

To begin with, this is an excellent study looking at a new diagnostic measure in a specific population and this will surely add to the existing literature and will help guide future studies.

I would suggest a few minor revisions (similar to what the reviewers have noted below)

Author should talk about the cost effectiveness of the study compared to angiogram and nuclear studyAuthors should note that retrospective nature of the study is a limitation. (please refer to reviewers concerns below about blinding in a retrospective study)Please review reviewer's suggestions below. Please submit your revised manuscript by Oct 25 2023 11:59PM. If you will need more time than this to complete your revisions, please reply to this message or contact the journal office at plosone@plos.org. Please include the following items when submitting your revised manuscript:A rebuttal letter that responds to each point raised by the academic editor and reviewer(s). You should upload this letter as a separate file labeled 'Response to Reviewers'.A marked-up copy of your manuscript that highlights changes made to the original version. You should upload this as a separate file labeled 'Revised Manuscript with Track Changes'.An unmarked version of your revised paper without tracked changes. You should upload this as a separate file labeled 'Manuscript'.If applicable, we recommend that you deposit your laboratory protocols in protocols.io to enhance the reproducibility of your results. Protocols.io assigns your protocol its own identifier (DOI) so that it can be cited independently in the future. For instructions see: https://journals.plos.org/plosone/s/submission-guidelines#loc-laboratory-protocols. Additionally, PLOS ONE offers an option for publishing peer-reviewed Lab Protocol articles, which describe protocols hosted on protocols.io. Read more information on sharing protocols at https://plos.org/protocols?utm_medium=editorial-email&utm_source=authorletters&utm_campaign=protocols.

We look forward to receiving your revised manuscript.

Kind regards,

Vikramaditya Samala Venkata

Academic Editor

PLOS ONE

Journal Requirements:

Reviewers' comments:

Reviewer's Responses to Questions

**Comments to the Author**

1. Is the manuscript technically sound, and do the data support the conclusions?

Reviewer #1: Yes

Reviewer #2: Yes

Reviewer #3: Yes

Reviewer #4: Yes

Reviewer #5: Yes

2. Has the statistical analysis been performed appropriately and rigorously? 

Reviewer #1: Yes

Reviewer #2: Yes

Reviewer #3: Yes

Reviewer #4: Yes

Reviewer #5: Yes

3. Have the authors made all data underlying the findings in their manuscript fully available?

Reviewer #1: Yes

Reviewer #2: Yes

Reviewer #3: Yes

Reviewer #4: Yes

Reviewer #5: Yes

4. Is the manuscript presented in an intelligible fashion and written in standard English?

Reviewer #1: Yes

Reviewer #2: Yes

Reviewer #3: Yes

Reviewer #4: Yes

Reviewer #5: Yes

5. Review Comments to the Author

Reviewer #1: I would first like to thank to give chance to review this study It is very good study looking at Asian population and safety , clinical effectivity of adenosine CMR stress test, there are few sentences that I recommended needs to be reworded/ restructured , otherwise the study / report clearly outlines the major versus minor complications noted during this test and clinical outcome

Reviewer #2: Well done study which seeks to ascertain the impact of Stress CMR in Asian population. It was a retrospective observational study but the authors mention that the cardiologists reviewing the data were blinded to the results, adverse patient events, etc. which is difficult in a retrospective study. Can authors please explain what precautions/ methodology was adopted to ensure blinding.

Reviewer #3: This is a retrospective observational study conducted in a Tertiary care hospital in Thailand. The authors have studied the safety and clinical impact of adenosine stress cardiac magnetic resonance(CMR) imaging MRI in patients with suspected or diagnosed coronary artery disease. The study is well done and reported methodically. Limited background data from Asia has been mentioned setting the context and need for the study. The authors have provided detailed background evidence and comparison to prior studies. They have collected a large sample size of 3768 patients from Southeast Asia which makes the data robust. Authors have reported on the demographics, clinical characteristics, hemodynamic variables(Blood pressure and heart rate) of the patients, and complications of the stress CMRI test.

The test is reported safe with only 4 major complications of acute pulmonary edema (0.44%). This is similar to the safety profile documented in prior studies.

Adenosine stress CMR had an overall impact on in 48% of the patients. 26% of patients got a new diagnosis ( CAD or non ischemic cardiac problems). 29% of the patient had a change in therapeutic plan( new medication or a procedure).

Overall the study has filled a gap in the knowledge about the safety and impact of adenosine stress CMR in the Asian population.

Reviewer #4: Considering the retrospective nature and region-specific, it is challenging to generalize the result however, other studies have shown a similar result.

It is important to mention the cost-effectiveness compared to nuclear study and CT coronary angiogram.

The table comparing the previous study is helpful. Mentioning the limitation at the end also gives more prospective for future study.

Overall, it is a well-done study.

Reviewer #5: Limited Applicability to Other Settings: This study was exclusively conducted at a single tertiary center in Thailand, which may limit the extent to which the results can be applied to diverse regions and populations with potentially distinct demographic compositions and healthcare systems. Acknowledging this constraint is essential, and the article should delve into the potential ramifications of this restriction.

Clarification of Patient Selection Criteria: The article would benefit from a more comprehensive elucidation of the criteria used to include or exclude patients in the study. For instance, it should explicitly define the criteria for excluding patients with contraindications to adenosine stress CMR.

Illustrating Clinical Impact: While the article highlights that stress CMR significantly impacted diagnosis and clinical management in 48% of patients, it could enhance reader comprehension by furnishing concrete examples of how CMR results influenced patient management. These practical instances would offer valuable insights into the real-world implications of the study's findings.

Contextualizing with Previous Research: While briefly alluding to studies conducted in Western countries, the article must include an opportunity for a more thorough comparative analysis. A more comprehensive examination of how this study's outcomes align with or diverge from prior research would give readers a richer contextual understanding of the findings.

6. PLOS authors have the option to publish the peer review history of their article (what does this mean?). If published, this will include your full peer review and any attached files.

Reviewer #1: **Yes: **Gurpreet Kaur Saini

Reviewer #2: No

Reviewer #3: No

Reviewer #4: **Yes: **Nihar Jena

Reviewer #5: **Yes: **Vishal Devarkonda

---

## [Author Response · Author response to Decision Letter 0]

20 Sep 2023

Editor ‘s comments

 To begin with, this is an excellent study looking at a new diagnostic measure in a specific population and this will surely add to the existing literature and will help guide future studies.

 We would like to express our sincere gratitude to the Editor for the dedicated time and effort invested in reviewing our manuscript. We highly value the insightful feedback provided by the Reviewers, which has undoubtedly contributed to the enhancement of our work. Below, we present our responses to the Review comments, addressing each point in a comprehensive manner. Your constructive guidance has been instrumental in refining our study, and we are truly appreciative of this invaluable contribution.

I would suggest a few minor revisions (similar to what the reviewers have noted below)

Author should talk about the cost effectiveness of the study compared to angiogram and nuclear study

 Thank you for the question. We agree with the editor that the cost-effectiveness analysis of CMR compared to angiogram and nuclear study is interesting. However, this is not a primary aim of our study. It requires several pieces of data that we were unable to obtain. Nevertheless, we have added this point to the limitation section. It is an idea for us to pursue further research in the future. 

Limitation section (page 13, line 304-306)

“Finally, our study did not include a cost-effective analysis of CMR compared to invasive coronary angiography or nuclear studies, as this was not our primary aim. However, this could serve as an opportunity for future research on this matter.”

Authors should note that retrospective nature of the study is a limitation. (please refer to reviewers concerns below about blinding in a retrospective study)

 Thank you for the comment. We have acknowledged this issue and added it to the limitation section (page 13 line 290-293)

Firstly, the retrospective nature of the study could introduce confounders that cannot be completely eliminated, and reviewing the medical records could potentially introduce some bias. However, the reviewers who assessed the patient information were blinded to the CMR results, which was the best approach we could take

Please review reviewer's suggestions below.

Done.

Review Comments to the Author

Reviewer #1: I would first like to thank to give chance to review this study It is very good study looking at Asian population and safety , clinical effectivity of adenosine CMR stress test, there are few sentences that I recommended needs to be reworded/ restructured , otherwise the study / report clearly outlines the major versus minor complications noted during this test and clinical outcome

 We sincerely thank the reviewer for dedicating their time and effort to review our article. The comments provided by the reviewer have greatly benefited our work, enhancing its clarity and overall quality. Our point-by-point response is provided below.

1. Abstract: we added the age of elderly as "elderly (ranging from 75 to 91 years)".

2. Page 2: Comment on sensitivity and specificity of adenosine stress CMR: We described this issue in the discussion section that demonstrated the accuracy data of adenosine stress CMR in our study.

Discussion section (page 10, line 225-229)

“Adenosine has been the most commonly used vasodilator for stress CMR and has been shown to exhibit better sensitivity and specificity compared to dipyridamole.(1) Our center has demonstrated good accuracy with adenosine stress CMR, achieving a sensitivity of 89.5% and a specificity of 78.6% compared to invasive coronary angiography.(2) These results are consistent with other published papers that also demonstrate good accuracy.(1)”

References

1) Hamon M, Fau G, Née G, Ehtisham J, Morello R, Hamon M. Meta-analysis of the diagnostic performance of stress perfusion cardiovascular magnetic resonance for detection of coronary artery disease. J Cardiovasc Magn Reson. 2010;12(1):29.

2) Krittayaphong R, Boonyasirinant T, Saiviroonporn P, Nakyen S, Thanapiboonpol P, Yindeengam A, et al. Myocardial perfusion cardiac magnetic resonance for the diagnosis of coronary artery disease: do we need rest images? Int J Cardiovasc Imaging. 2009;25 Suppl 1:139-48.

3. Introduction: We specify clinical impact of stress CMR in patients with known or suspected CAD.

Introduction section (page 3, line 64-66)

“Stress CMR has demonstrated a significant impact on the diagnosis and management of a large patient population, as evidenced by The EuroCMR registry, which included more than 9,500 patients suspected of CAD or suspected ischemia in known CAD who underwent stress tests.(1)”

Reference

1) Bruder O, Wagner A, Lombardi M, Schwitter J, van Rossum A, Pilz G, et al. European cardiovascular magnetic resonance (EuroCMR) registry – multinational results from 57 centers in 15 countries. Journal of Cardiovascular Magnetic Resonance. 2013;15(1):9.

4. Method: A question regarding discontinue of adenosine infusion.

 For our CMR protocol, we allow patients to request the discontinuation of adenosine, even though we instruct them to be aware of the adverse effects of adenosine. Very few of them had severe symptom such as dyspnea and chest pain. Despite allowing patients to request the discontinuation of adenosine, there was no record of a patient being unable to tolerate it to the extent that the test had to be stopped.

5. Results: We have revised two sentences regarding characteristics of four patients who had acute pulmonary edema after CMR.

Result section (page 8, line 179-183)

“All four patients, aged over 75 (ranging from 75 to 91 years), required hospital observation or admission for further management. They had previously been diagnosed with heart failure, classified as New York Heart Association (NYHA) class II-III, with two having reduced left ventricular ejection fraction and two having preserved left ventricular ejection fraction.”

6. Results: We clarify local complications at intravenous access site.

Results section (page 8, line 187-188)

“Local complications at intravenous access sites, such as small hematoma, edema, or phlebitis (n=7)”

7. Discussion: We added reasons that acute pulmonary edema has occurred only in patients with heart failure.

Discussion section (page 10-11, line 237-241)

“Four patients with acute pulmonary edema were reported, all of whom were elderly and had symptomatic heart failure. It is known that a comprehensive stress CMR protocol requires patient cooperation, as they have to remain supine in a magnet for at least 30 minutes and repeatedly hold their breath. Patients with symptomatic heart failure, especially those with fluid retention, are at a high risk for a heart failure event during or after CMR.”

Reviewer #2: Well done study which seeks to ascertain the impact of Stress CMR in Asian population. It was a retrospective observational study but the authors mention that the cardiologists reviewing the data were blinded to the results, adverse patient events, etc. which is difficult in a retrospective study. Can authors please explain what precautions/ methodology was adopted to ensure blinding.

 Thank you for your comment; this is indeed an important issue. We apologize for any confusion. In our study, two cardiologists reviewed patient information, including baseline characteristics, indications for CMR, adverse events during CMR studies, and changes in management after CMR. Importantly, they did so without access to the CMR results, whether they were positive or negative for ischemia or positive or negative for LGE. We are aware that this approach may introduce bias due to the retrospective nature of the study. Nevertheless, we made every effort to minimize this bias and believe that this was the most suitable approach for a retrospective study. Furthermore, we have revised this section and acknowledged this limitation in the limitations section.

Method section (page 6, line 137-139)

“Two cardiologists, who were blinded to the CMR results, reviewed patient information, including baseline characteristics, indications for CMR, adverse events during CMR studies, and changes in management after CMR. They independently assessed the clinical impact of each stress CMR by reviewing electronic medical records up to the next outpatient visit with the ordering provider. A “completely new diagnosis” was defined as a diagnosis occurring only if it was previously unknown to the referring physician.”

Limitation section (page 13, line 290-293)

Firstly, the retrospective nature of the study could introduce confounders that cannot be completely eliminated, and reviewing the medical records could potentially introduce some bias. However, the reviewers who assessed the patient information were blinded to the CMR results, which was the best approach we could take

Reviewer #3: This is a retrospective observational study conducted in a Tertiary care hospital in Thailand. The authors have studied the safety and clinical impact of adenosine stress cardiac magnetic resonance(CMR) imaging MRI in patients with suspected or diagnosed coronary artery disease. The study is well done and reported methodically. Limited background data from Asia has been mentioned setting the context and need for the study. The authors have provided detailed background evidence and comparison to prior studies. They have collected a large sample size of 3768 patients from Southeast Asia which makes the data robust. Authors have reported on the demographics, clinical characteristics, hemodynamic variables(Blood pressure and heart rate) of the patients, and complications of the stress CMRI test.

The test is reported safe with only 4 major complications of acute pulmonary edema (0.44%). This is similar to the safety profile documented in prior studies.

Adenosine stress CMR had an overall impact on in 48% of the patients. 26% of patients got a new diagnosis (CAD or non-ischemic cardiac problems). 29% of the patient had a change in therapeutic plan( new medication or a procedure).

Overall, the study has filled a gap in the knowledge about the safety and impact of adenosine stress CMR in the Asian population.

 We thank the reviewer for their dedicated time and effort in reviewing our study, and we greatly appreciate their compliments suggesting that our study will be beneficial for patient management.

Reviewer #4: Considering the retrospective nature and region-specific, it is challenging to generalize the result however, other studies have shown a similar result.

It is important to mention the cost-effectiveness compared to nuclear study and CT coronary angiogram.

The table comparing the previous study is helpful. Mentioning the limitation at the end also gives more prospective for future study.

Overall, it is a well-done study.

 Thank you for the question. We agree with the editor that the cost-effectiveness analysis of CMR compared to angiogram and nuclear study is interesting. However, this is not a primary aim of our study. It requires several pieces of data that we were unable to obtain. Nevertheless, we have added this point to the limitation section. It is an idea for us to pursue further research in the future. 

Limitation section (page 13, line 304-306)

“Finally, our study did not include a cost-effective analysis of CMR compared to invasive coronary angiography or nuclear studies, as this was not our primary aim. However, this could serve as an opportunity for future research on this matter.”

Reviewer #5: 

1. Limited Applicability to Other Settings: This study was exclusively conducted at a single tertiary center in Thailand, which may limit the extent to which the results can be applied to diverse regions and populations with potentially distinct demographic compositions and healthcare systems. Acknowledging this constraint is essential, and the article should delve into the potential ramifications of this restriction.

 Thank you for your comment. We acknowledge the limitation that our study was conducted in a tertiary center in Thailand, as we stated in the limitations section. However, our study's focus was on Asia, where there is limited data available regarding the safety of CMR and its impact on patient management. Additionally, it is worth noting that, in general, CMR centers tend to be tertiary centers, especially in Asia, like ours. We believe that our study represents patients in typical tertiary centers offering CMR services in Asia.

2. Clarification of Patient Selection Criteria: The article would benefit from a more comprehensive elucidation of the criteria used to include or exclude patients in the study. For instance, it should explicitly define the criteria for excluding patients with contraindications to adenosine stress CMR.

 Thank you for the valuable comments. In our study, we retrospectively included consecutive patients aged >18 years who were referred for adenosine stress CMR. This study reflects the real world and includes all patients who underwent a specific CMR protocol at our center. We did not specify the exclusion criteria since patients with contraindications for CMR were not included in the study because they did not undergo the scan. However, we thank the reviewer once again for bringing up this point.

3. Illustrating Clinical Impact: While the article highlights that stress CMR significantly impacted diagnosis and clinical management in 48% of patients, it could enhance reader comprehension by furnishing concrete examples of how CMR results influenced patient management. These practical instances would offer valuable insights into the real-world implications of the study's findings.

 As the reviewer stated, our study highlights the safety and clinical impact of adenosine stress CMR for patient management. We believe this data will assist clinicians in ensuring that adenosine CMR is indeed very safe and has a clinical impact on patient management. This will also enhance the confidence of clinicians and their patients in the benefits of CMR. Such confidence will promote CMR as a one-stop service offering accurate diagnostic testing with an excellent safety profile and significant clinical impact. We have emphasized this point in the discussion section.

Discussion section (page 13, line 284-289)

 “For applicability, our data highlights the safety and clinical impact of adenosine stress CMR for patient management. This data will assist clinicians in ensuring that adenosine CMR is indeed very safe and has a clinical impact on patient management. This will also enhance the confidence of clinicians and their patients in the benefits of CMR. Such confidence will promote CMR as a one-stop service offering accurate diagnostic testing with an excellent safety profile and significant clinical impact.”

4. Contextualizing with Previous Research: While briefly alluding to studies conducted in Western countries, the article must include an opportunity for a more thorough comparative analysis. A more comprehensive examination of how this study's outcomes align with or diverge from prior research would give readers a richer contextual understanding of the findings. 

 Thank you for the valuable comment. This enhances our study to be much better. As the reviewer suggested, we added a discussion regarding our results and compared them with other published articles.

Discussion section (page 12, line 272-283)

 “Table 4 presents a summary of patients' profiles, complications, and the clinical impact of stress CMR in our study, compared with previously published data.(1-5) Our data demonstrated some degree of similarity in patient profiles, such as having 28% with known CAD, consistent with Menadas et al. and McGraw et al.(2,3) Notably, our study had a lower proportion of male participants compared to most studies, as male patients may have a higher pretest probability of obstructive CAD and may undergo invasive coronary angiography rather than CMR. Regarding complications, our study and all studies had quite similar very low rates of major complications (<0.3% in all studies), showing a consistent safety profile of stress CMR, while there were some differences in baseline characteristics. As for the clinical impact of CMR, although there were differences in the definitions of impact and/or therapeutic consequences, all studies also showed a consistent and significant impact of CMR. Overall, we have added data from Asia showing that CMR is safe and has a clinical impact compared to studies from Western countries.”

References

1) Bruder O, Wagner A, Lombardi M, Schwitter J, van Rossum A, Pilz G, et al. European cardiovascular magnetic resonance (EuroCMR) registry – multi national results from 57 centers in 15 countries. Journal of Cardiovascular Magnetic Resonance. 2013;15(1):9.

2) McGraw S, Romano S, Jue J, Bauml MA, Chung J, Farzaneh-Far A. Impact of Stress Cardiac Magnetic Resonance Imaging on Clinical Care. Am J Cardiol. 2016;118(6):924-9.

3) Monmeneu Menadas JV, Lopez-Lereu MP, Estornell Erill J, Garcia Gonzalez P, Igual Muñoz B, Maceira Gonzalez A. Pharmacological stress cardiovascular magnetic resonance: feasibility and safety in a large multicentre prospective registry. European Heart Journal - Cardiovascular Imaging. 2015;17(3):308-15.

4) Tsang KH, Chan WS, Shiu CK, Chan MK. The safety and tolerability of adenosine as a pharmacological stressor in stress perfusion cardiac magnetic resonance imaging in the Chinese population. Hong Kong Med J. 2015;21(6):524-7.

5) Raj V, Pudhiavan A, Hrishikesh VJ, Ali A, Kothari R. Safety profile of adenosine stress cardiac MRI in a tertiary hospital in India. Indian J Radiol Imaging. 2020;30(4):459-64.

---

## [Editor Report · Decision Letter 1]

3 Oct 2023

Safety and Clinical Impact of Adenosine Stress Perfusion Cardiac Magnetic Resonance in Asian Patients with Known or Suspected Coronary Artery Disease

PONE-D-23-23709R1

Dear Dr. Krittayaphong,

We’re pleased to inform you that your manuscript has been judged scientifically suitable for publication and will be formally accepted for publication once it meets all outstanding technical requirements.

Kind regards,

Vikramaditya Samala Venkata

Academic Editor

PLOS ONE
---

## [Editor Report · Acceptance letter]

5 Oct 2023

PONE-D-23-23709R1 

Safety and Clinical Impact of Adenosine Stress Perfusion Cardiac Magnetic Resonance in Asian Patients with Known or Suspected Coronary Artery Disease 

Dear Dr. Krittayaphong:

I'm pleased to inform you that your manuscript has been deemed suitable for publication in PLOS ONE. Congratulations! Your manuscript is now with our production department. 

Kind regards, 

on behalf of

Dr. Vikramaditya Samala Venkata 

Academic Editor

PLOS ONE